# Globally Gated Deep Linear Networks

**Qianyi Li**[1]     **Haim Sompolinsky**[2,3]
[1]Biophysics Graduate Program, Harvard University
[2]Center for Brain Science, Harvard University
[3]Edmond and Lily Safra Center for Brain Sciences, Hebrew University
qianyi_li@g.harvard.edu,  hsompolinsky@mcb.harvard.edu,  haim@fiz.huji.ac.il

## Abstract

Recently proposed Gated Linear Networks (GLNs) present a tractable nonlinear network architecture, and exhibit interesting capabilities such as learning with local error signals and reduced forgetting in sequential learning. In this work, we introduce a novel gating architecture, named Globally Gated Deep Linear Networks (GGDLNs) where gating units are shared among all processing units in each layer, thereby decoupling the architectures of the nonlinear but unlearned gating and the learned linear processing motifs. We derive exact equations for the generalization properties of Bayesian Learning in these networks in the finite-width thermodynamic limit, defined by $N, P \to \infty$ while $P/N = O(1)$ where $N$ and $P$ are the hidden layers' width and size of training data sets respectfully. We find that the statistics of the network predictor can be expressed in terms of kernels that undergo shape renormalization through a data-dependent order parameter matrix compared to the infinite-width Gaussian Process (GP) kernels. Our theory accurately captures the behavior of finite width GGDLNs trained with gradient descent (GD) dynamics. We show that kernel shape renormalization gives rise to rich generalization properties w.r.t. network width, depth and $L_2$ regularization amplitude. Interestingly, networks with a large number of gating units behave similarly to standard ReLU architectures. Although gating units in the model do not participate in supervised learning, we show the utility of unsupervised learning of the gating parameters. Additionally, our theory allows the evaluation of the network's ability for learning multiple tasks by incorporating task-relevant information into the gating units. In summary, our work is the first exact theoretical solution of learning in a family of nonlinear networks with finite width. The rich and diverse behavior of the GGDLNs suggests that they are helpful analytically tractable models of learning single and multiple tasks, in finite-width nonlinear deep networks.

## 1   Introduction

Despite the recent advances in machine learning, theoretical understanding of how machine learning algorithms work is very limited. Many current theoretical approaches study infinitely wide networks [1, 2, 3], where the input-output relation is equivalent to a Gaussian Process (GP) in function space with a covariance matrix defined by a GP kernel. However, this GP limit holds when the network width approaches infinity while the size of the training data remains finite, severely limiting its applicability to realistic conditions. Another line of work focuses on finite-width deep linear neural networks (DLNNs)[4, 5, 6], while applicable in a wider regime, the generalization behavior of linear networks are very limited, and the bias contribution always remains constant with network parameters [4], which fails to capture the behavior of generalization performance in general nonlinear networks. Therefore, a tractable nonlinear network architecture is in need for theoretically probing into the diverse generalization behavior of general nonlinear networks.

36th Conference on Neural Information Processing Systems (NeurIPS 2022).

Recently proposed Gated Linear Networks (GLNs) present a tractable nonlinear network architecture [7, 8, 9], with capabilities such as learning with local error signals and mitigating catastrophic forgetting in sequential learning. Inspired by these recent advances in GLNs, we propose Globally Gated Deep Linear Networks (GGDLNs) as a simplified GLN structure that preserves the nonlinear property of general GLNs, the decoupling of fixed nonlinear gating from learned linear processing units, and the ability to separate the processing of multiple tasks using the gating units. Our GGDLN structure is different from previous GLNs in several ways. First, the gating units are shared across hidden layer units and different layers while in previous work each unit has its own set of gatings [10, 8, 9]. Second, we define global learning objective instead of local errors [8, 9]. These simplifications allow us to obtain direct analytical expressions of memory capacity and exact generalization error of these networks for arbitrary training and testing data, providing quantitative insight into the effect of learning in nonlinear networks, as opposed to studies of generalization bounds [10], expressivity estimates [9, 11], and indirect quantities relevant to generalization such as the implicit bias of the network [12]. Furthermore, the kernel expression of the predictor statistics we propose in this work also allow us to make qualitative explanations of the generalization and how it's related to data structure and network representation for single and multiple tasks.

First, we introduce the architecture of our GGDLNs and analyze its memory capacity. We then derive our theory for generalization properties of GGDLNs, and make qualitative connections between the generalization behavior and the relation between the renormalization matrix and task structure. Second, we apply our theory to GGDLNs performing multiple tasks, focusing on two scenarios where tasks are either defined by different input statistics or different output labels on the same inputs. While the effect of kernel renormalization is different in the two cases, we find that for fixed gating functions, de-correlation between tasks always improves generalization.

## 2 Globally gated deep linear networks

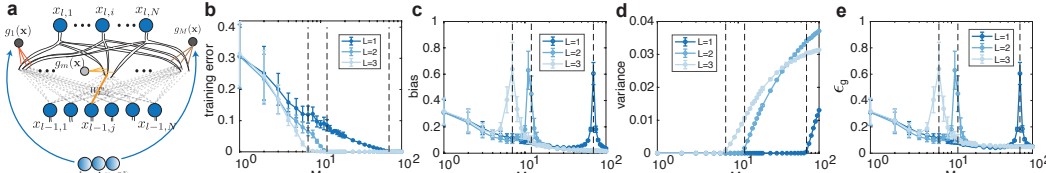

Figure 1: Globally gated deep linear networks. (**a**) Structure of GGDLNs, each neuron in the hidden layer has $M$ dendrites, each with a different input-dependent gating $g_m(\mathbf{x})$ which is fixed during training, the $M$ gatings are shared across neurons in the hidden layer. The $m$-th dendritic branch of the $i$-th neuron in layer $l$ connects to neuron $j$ in the previous layer with weight $W_{l,ij}^m$(shown in orange). (**b**) Training error of networks with 1-3 hidden layers in the GP limit as a function of $M$ evaluated on a noisy ReLU teacher task, training error goes to zero at network capacity (black dashed lines). (**c-e**) Bias, variance and generalization error of the same network and task as (b). Bias and generalization error diverges, variance generalization becomes nonzero at network capacity (black dashed line). See Appendix C.1 for detailed parameters.

In GGDLNs, the network input-output relation is defined as follows,

$$f(\mathbf{x}) = \frac{1}{\sqrt{NM}} \sum_{i=1}^{N} \sum_{m=1}^{M} a_{m,i} x_{L,i} g_m(\mathbf{x}), \ x_{l,i} = \begin{cases} \frac{1}{\sqrt{N_0 M}} \sum_{j=1}^{N_0} \sum_{m=1}^{M} W_{l,ij}^m g_m(\mathbf{x}) x_{l-1,j} & l > 1 \\ \frac{1}{\sqrt{N_0}} \sum_{j=1}^{N_0} W_{l,ij} x_{l-1,j} & l = 1 \end{cases}$$

(1)

where $\mathbf{x}_0 = \mathbf{x}$ is the input, $N$ is the hidden layer width, $M$ is the number of gating units in each layer, and $N_0$ is the input dimension. Each neuron in every layer has $M$ dendrites, each with an input-dependent global gating $g_m(\mathbf{x})$ shared across all neurons. The $m$-th dendritic branch of neuron $i$ in the $L$-th hidden layer connects to neurons in the previous layer with a dendrite-specific weight vector $\mathbf{W}_{L,i}^m$ (or with readout weight vector $\mathbf{a}_m$ for the output neuron), as shown in Fig.1 (a). Note that although the gatings are fixed during learning, changes in the weights affect how these gatings act on the hidden layer activations, and it is interesting to understand how the learned task interacts with these gating operations. Since adding gatings at the input layer is equivalent to expanding the

input dimension and replacing $x_j$ by $x_j g_m(\mathbf{x})$, and learning does not affect how the gatings interact with the input, *we do not add gatings at the input layer for simplicity.*

**Memory Capacity:** Memory capacity refers to the maximum number of random (or generic) input-output examples for which there exists a set of parameters such that the network achieves zero training error (here we consider the mean squared error, MSE). By definition, it is irrespective of the learning algorithm. The capacity bounds of deep nonlinear networks has been extensively studied in many recent works [10, 13, 14]. To calculate the capacity of GGDLNs, note that the input-output relation given by Eq.1 can be alternatively expressed as $f(\mathbf{x}) = \sum_{m_1,\cdots,m_L,j} W^{\text{eff}}_{m_1,\cdots,m_L,j} x^{\text{eff}}_{m_1,\cdots,m_L,j}$, which is a linear combination of the effective input $x^{\text{eff}}_{m_1,\cdots,m_L,j} = g_{m_1}(\mathbf{x}) g_{m_2}(\mathbf{x}) \cdots g_{m_L}(\mathbf{x}) x_j$ ($m_l = 1, \cdots, M; j = 1, \cdots, N_0$), with some effective weights $\mathbf{W}^{\text{eff}}$ which is a complicated function of $\mathbf{a}$ and $\mathbf{W}$. Here $m_l$ is the index of the gatings in the $l$-th layer. As the gating units are shared across layers, the effective input $\mathbf{x}^{\text{eff}}$ has $N_0 \binom{M+L-1}{L}$ independent dimensions. This combinatorial term represents the number of possible combinations of L gatings selected from M total number of gatings. Assuming $N \gg M^L$ such that the effective weight $W^{\text{eff}}_{m_1,\cdots,m_L,j}$ can take any desired value in the $N_0 M^L$ dimensional whole space, the problem of finding $\mathbf{W}^{\text{eff}}$ with zero training error is equivalent to a linear regression problem with input $\mathbf{x}^{\text{eff}}$ and the target outputs. Therefore, the capacity is equivalent to the number of independent input dimensions, given by $P \leq N_0 \binom{M+L-1}{L}$. The above capacity is verified by Fig.1(b), where the training error becomes nonzero above the memory capacity. The generalization behavior also changes drastically at network capacity (Fig.1(c-e)), where generalization error and its bias contribution diverge, and the variance contribution shrinks to 0 (see detailed calculation in the next paragraph and Appendix A.3). This double descent property of the generalization error is similar to previously studied in linear and nonlinear networks. Furthermore, although the output of the network is a linear function of the effective input $\mathbf{x}^{\text{eff}}$, due to the multiplicative nature of the network weights and the gatings, learning in GGDLNs is highly nonlinear and the space of solution for $\mathbf{W}$ and $\mathbf{a}$ is highly nontrivial, and the network exhibit properties unique to nonlinear networks, as we will show in the following sections.

**Posterior distribution of network weights:** We consider a Bayesian network setup, where the network weights are random variables whose statistics are determined by the training data and network parameters, instead of deterministic variables. This probabilistic approach enables us to study the properties of *the entire solution space* instead of a single solution which may be heavily initialization dependent. We consider the posterior distribution of the network weights induced by learning with a Gaussian prior [15, 16, 17, 18], given by

$$P(\boldsymbol{\Theta}) = Z^{-1} \exp(-\frac{1}{2T} \sum_{\mu=1}^{P} (f(\mathbf{x}^{\mu}, \boldsymbol{\Theta}) - Y^{\mu})^2 - \frac{1}{2\sigma^2} \boldsymbol{\Theta}^{\top} \boldsymbol{\Theta}) \tag{2}$$

where $Z$ is the partition function $Z = \int d\boldsymbol{\Theta} P(\boldsymbol{\Theta})$. The first term in the exponent is the MSE of the network outputs on a set of $P$ training data points $\mathbf{x}^{\mu}$ from their target outputs $Y^{\mu}$, and the second term is a Gaussian prior on the network parameters $\boldsymbol{\Theta} = \{\mathbf{W}, \mathbf{a}\}$ with amplitude $\sigma^{-2}$. In this work we focus on the $T \to 0$ limit where the first term dominates. Below the network capacity, the distribution of $\boldsymbol{\Theta}$ concentrates onto the solution space that yields zero training error, the Gaussian prior then biases the solution space towards weights with smaller $L_2$ norms. The fundamental properties of the system can be derived from the partition function. As the distribution is quadratic in the readout weights $a_{m,i}$, it is straightforward to integrate them out, which yields

$$Z = \int d\mathbf{W} \exp[-\frac{1}{2\sigma^2} \text{Tr}(\mathbf{W}^{\top}\mathbf{W}) + \frac{1}{2}\mathbf{Y}^{\top}\mathbf{K}_L(\mathbf{W})^{-1}\mathbf{Y} + \frac{1}{2}\log\det(\mathbf{K}_L(\mathbf{W}))] \tag{3}$$

where $\mathbf{W}$ denotes all the remaining weights in the network, and $\mathbf{K}_L(\mathbf{W})$ is the $\mathbf{W}$ dependent $P \times P$ kernel on the training data, defined as $K_L^{\mu\nu}(\mathbf{W}) = (\frac{\sigma^2}{M}\mathbf{g}(\mathbf{x}^{\mu})^{\top}\mathbf{g}(\mathbf{x}^{\nu}))(\frac{1}{N}\mathbf{x}_L^{\mu}(\mathbf{W})^{\top}\mathbf{x}_L^{\nu}(\mathbf{W}))$.

**Generalization in infinitely wide GGDLNs:** It is well known that in infinitely wide networks where $N \to \infty$ while $P$ remains finite (also referred to as the GP limit), $\mathbf{K}_L(\mathbf{W})$ is self-averaging and does not depend on the specific realization of $\mathbf{W}$. It can therefore be replaced by the GP kernel defined as $\langle \mathbf{K}_L(\mathbf{W}) \rangle_{\mathbf{W}}$ where $\mathbf{W} \sim \mathcal{N}(0, \sigma^2)$ [2]. For GGDLNs, the GP kernel for a pair of arbitrary data $\mathbf{x}$ and $\mathbf{y}$ is given by $K_{GP}(\mathbf{x}, \mathbf{y}) = (\frac{\sigma^2}{M}\mathbf{g}(\mathbf{x})^{\top}\mathbf{g}(\mathbf{y}))^L K_0(\mathbf{x}, \mathbf{y})$, where $K_0(\mathbf{x}, \mathbf{y}) = \frac{\sigma^2}{N_0}\mathbf{x}^{\top}\mathbf{y}$. We denote the $P \times P$ kernel data *matrix* as $\mathbf{K}_{GP}$ where $K_{GP}^{\mu\nu} = K_{GP}(\mathbf{x}^{\mu}, \mathbf{x}^{\nu})$, and the input kernel matrix on training data as $\mathbf{K}_0$ where $K_0^{\mu\nu} = K_0(\mathbf{x}^{\mu}, \mathbf{x}^{\nu})$.

Generalization error is measured by MSE including the bias and the variance contributions, $\epsilon_g = \underbrace{(\langle f(\mathbf{x})\rangle_\Theta - y(\mathbf{x}))^2}_{\text{bias}} + \underbrace{\langle \delta f(\mathbf{x})^2 \rangle_\Theta}_{\text{variance}}$, which depends on the first and second order statistics of the predictor. In the GP limit, we have

$$\langle f(\mathbf{x})\rangle = \mathbf{k}_{GP}(\mathbf{x})^\top \mathbf{K}_{GP}^{-1}\mathbf{Y}, \ \langle \delta f(\mathbf{x})^2 \rangle = \mathbf{K}_{GP}(\mathbf{x},\mathbf{x}) - \mathbf{k}_{GP}(\mathbf{x})^\top \mathbf{K}_{GP}^{-1}\mathbf{k}_{GP}(\mathbf{x}) \tag{4}$$

where $k_{GP}^\mu(\mathbf{x}) = K_{GP}(\mathbf{x},\mathbf{x}^\mu)$. Note that the rank of $\mathbf{K}_{GP}$ is the same as the capacity of the network, and the kernel matrix becomes singular as $P$ approaches its capacity (the interpolation threshold), which results in nonzero training error, diverging bias and vanishing variance contribution to the generalization error (Fig.1 (b-e)). The singularity of the kernel at the interpolation threshold holds also for finite width networks, and similar diverging bias and vanishing variance are seen in our finite width theory below (Section 3 ) and are confirmed by simulation of networks trained with GD (see Appendix B.1, [19]).

## 3   Kernel shape renormalization theory in finite-width GLNs

We now address the finite width thermodynamic limit, where $P, N \to \infty$ but $P/N \sim \mathcal{O}(1)$, $M, L \sim \mathcal{O}(1)$. In this limit, calculating the statistics of the network predictor requires integration over $\mathbf{W}$ in Eq.3 . To do so, we apply the previous method of Back-propagating Kernel Renormalization [4] (see Appendix A) to GGDLNs. The partition function for a single hidden layer network is given by $Z = \exp(-H_1)$, where the Hamiltonian $H_1$ is given by

$$H_1 = \frac{1}{2}\mathbf{Y}^\top \tilde{\mathbf{K}}_1^{-1}\mathbf{Y} + \frac{1}{2}\log\det(\tilde{\mathbf{K}}_1) - \frac{N}{2}\log\det\mathbf{U}_1 + \frac{1}{2\sigma^2}N\text{Tr}(\mathbf{U}_1)$$
$$\tilde{K}_1^{\mu\nu} = (\frac{1}{M}\mathbf{g}(\mathbf{x}^\mu)^\top \mathbf{U}_1 \mathbf{g}(\mathbf{x}^\nu))K_0^{\mu\nu} \tag{5}$$

Comparing the matrix $\tilde{\mathbf{K}}_1$ to $\mathbf{K}_{GP}$, we note that the GP kernel is renormalized by an an $M \times M$ matrix order parameter $\mathbf{U}_1$. This order paramter satisfies the self-consistent equation

$$\mathbf{U}_1 = I - \frac{1}{NM}\mathbf{U}_1^{1/2}\mathbf{g}^\top[\tilde{\mathbf{K}}_1^{-1} \circ \mathbf{K}_0]\mathbf{g}\mathbf{U}_1^{1/2} + \frac{1}{NM}\mathbf{U}_1^{1/2}\mathbf{g}^\top[\tilde{\mathbf{K}}_1^{-1}\mathbf{Y}\mathbf{Y}^\top\tilde{\mathbf{K}}_1^{-1} \circ \mathbf{K}_0]\mathbf{g}\mathbf{U}_1^{1/2} \tag{6}$$

where $\circ$ denotes element-wise multiplication. In the linear case (which corresponds to $M = 1$), the GP kernel is renormalized by a scalar factor. In the $M > 1$ case, the effect of renormalization is more drastic as it changes that not only the amplitude but also the shape of the kernel. The renormalization matrix has an interesting physical interpretation that relates it to the readout weights $\mathbf{a}$ of GGDLNs,

$$U_1^{mn} = \langle \frac{1}{N}\sum_{i=1}^N a_{m,i}a_{n,i}\rangle \tag{7}$$

The calculation can be extended to multiple layers with a new order parameter introduced for each layer (see Appendix A). The predictor statistics for a input $\mathbf{x}$ can be expressed in terms of the renormalized kernels, for a network with $L = 1$

$$\langle f(\mathbf{x})\rangle_\Theta = \tilde{\mathbf{k}}_1(\mathbf{x})^\top \tilde{\mathbf{K}}_1^{-1}\mathbf{Y}, \ \langle \delta f(\mathbf{x})^2 \rangle_\Theta = \tilde{K}_1(\mathbf{x},\mathbf{x}) - \tilde{\mathbf{k}}_1(\mathbf{x})^\top \tilde{\mathbf{K}}_1^{-1}\tilde{\mathbf{k}}_1(\mathbf{x}) \tag{8}$$

where $\tilde{K}_1(\mathbf{x},\mathbf{y}) = (\frac{1}{M}\mathbf{g}(\mathbf{x})^\top\mathbf{U}_1\mathbf{g}(\mathbf{y}))K_0(\mathbf{x},\mathbf{y})$ denotes the renormalized kernel *function* for two arbitrary inputs $\mathbf{x}$ and $\mathbf{y}$, $\tilde{\mathbf{K}}_1$ denotes the $P \times P$ renormalized kernel *matrix* on the training data, and $\tilde{\mathbf{k}}_1(\mathbf{x})$ is a $P$-dimensional vector, $\tilde{k}_1^\mu(\mathbf{x}) = K(\mathbf{x},\mathbf{x}^\mu)$. The kernel renormalization in GGDLNs changes the shape of the kernel through the data dependent $\mathbf{U}_1$, reflecting the nonlinear property of the network, and resulting in more complex behavior of predictor statistics relative to the linear networks, as shown in Section 4. Our theory describes the properties of the posterior distribution of the network weights induced at equilibrium by Langevin dynamics with the MSE cost function and the Gaussian prior [4, 20, 21, 22]. Simulating this dynamics agrees remarkably well with the simulation (see Appendix B.2). Although our theoretical results do not directly describe the solutions obtained by running gradient descent (GD) dynamics on the training error, it is interesting to ask to what extent the predicted behaviors of our theory are also exhibited by GD dynamics of the same network architectures, as GD-based learning is more widely used. We will compare our theoretical results with numerics of GD dynamics throughout the paper. We consider the case where

the network is initialized with Gaussian i.i.d. weights with variance $\sigma^2$, and the mean and variance of the predictors are evaluated across multiple initializations (see Appendix C.5 for details). As we will show, our theory makes accurate qualitative predictions for GD dynamics in all examples in this paper, in the sense that while the exact values may not match, the general trend of how generalization or representation varies with different parameters in different regimes are very similar.

## 4 Generalization

For linear networks the generalization error depends on $N, \sigma^2$ and $L$ through the variance only, while the mean predictor always assumes the same value as in the GP limit [4]. This is because the scalar kernel renormalization of $\tilde{\mathbf{k}}_1(\mathbf{x})$ is cancelled out in the mean predictor by the renormalization of the inverse kernel $\tilde{\mathbf{K}}_1^{-1}$. In contrast, for GGDLNs the mean predictor and hence the error bias also change with these network parameters due to the matrix nature of the kernel renormalization (Eq.8) . Below we investigate in detail how matrix renormalization of the kernel affects the generalization behavior (especially the bias term) of the network.

### 4.1 Networks with single hidden layer

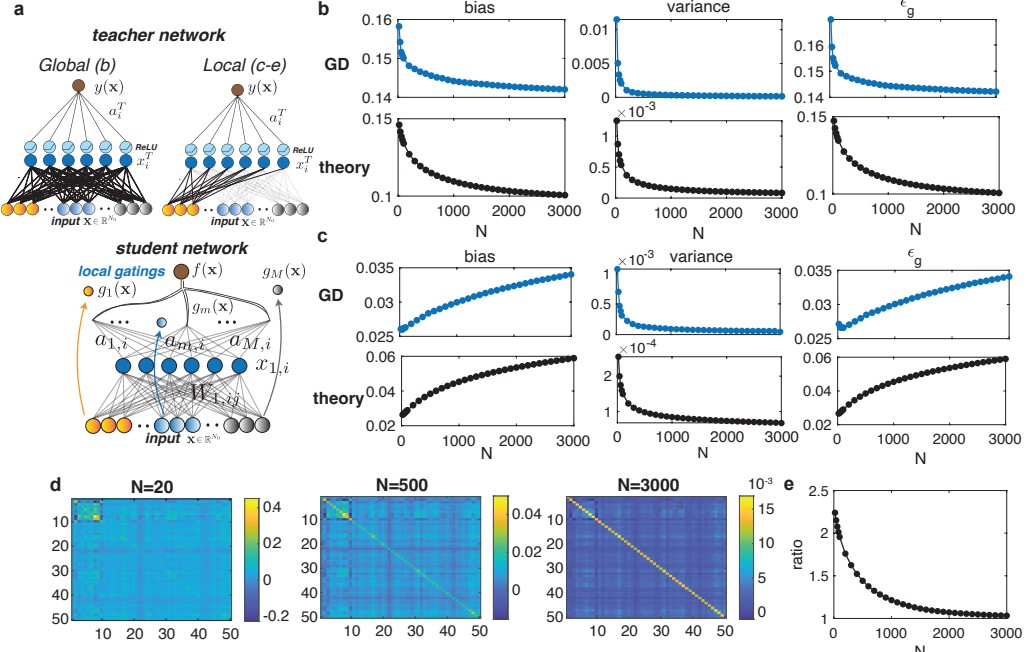

Figure 2: Dependence of generalization error on network width for a ReLU teacher task. (**a**)*Top:* The ReLU teacher network, the input $\mathbf{x}$ is divided into 5 subsets of input dimensions, the input layer weights either assume same order of magnitude across different input dimensions (left, (b)), or assume larger amplitudes for one subset of input dimensions-the preferred inputs (right, bold connections to a subset of input neurons,(c-e)). *Bottom:* The student network is a GGDLN with one hidden layer and gatings with localized receptive fields: each gating is connected to only a subset of input dimensions. (**b**) Bias, variance and generalization error decreases as a function of $N$ for a regular ReLU teacher, theory agrees qualitatively well with GD dynamics. (**c**) Bias and generalization error increases as a function of $N$ for ReLU teacher with preferred inputs. (**d**) The renormalization matrix $\mathbf{U}_1$ for different network widths for the teacher with preferred inputs. The first $10 \times 10$ block corresponds to the gatings with the same receptive field as the teacher's preferred inputs, and is amplified for small $N$. (**e**) The ratio of the average amplitude of the first $10 \times 10$ block relative to the average amplitude of the other four $10 \times 10$ diagonal blocks decreases as a function of $N$.

**Feature selection in finite-width networks:** Unlike in DLNs, the bias term in GGDLNs depends on $N$, exhibiting different dependence in different parameter regimes. This dependence also varies with

the choice of the gating functions. In Fig.2 we consider a student-teacher learning task, commonly used for evaluating and understanding neural network performance[23, 24, 25, 26]. We present results of learning a ReLU teacher task in GGDLNs with gatings that have *localized receptive fields* (i.e., the activation of each gating unit depends on only a subset of input dimensions, the receptive field of all gating units tile the $N_0$ input dimensions, as shown in Fig.2(a) bottom), where the student GGDLN is required to learn the input-output relation of a given ReLU teacher. For a ReLU teacher with a single fully connected hidden layer (Fig.2(a) top left), gatings with different receptive fields are of equal importance, hence the renormalization does not play a beneficial functional role, and the infinitely wide network performs better than finite $N$. As shown in Fig.2(b), bias, variance and generalization error all decrease with $N$. For a 'local' ReLU teacher with larger input weights for one subset of input components (the preferred inputs, Fig.2(a) top right), renormalization improves task performance by the selective increase of the elements in $\mathbf{U}_1$ that correspond to gating units whose receptive fields overlap the teacher's preferred inputs (Fig.2(d&e)). Hence, narrower networks (with a stronger renormalization) generalize better, and both the bias and the generalization error increase with $N$ (Fig.2(c)). More generally, the input can represent a set of fixed features of the data, and the 'local' teacher generate labels depending on a subset of the features. Therefore, networks with finite width are able to select the relevant set of features by adjusting the amplitude in the renormalization matrix $\mathbf{U}_1$ to assign the gating units with different importance for the task, while in the GP limit the network always assigns equal importance to all the gating units.

To summarize, our theory not only captures the more complex behavior of generalization (especially bias) as a function of network width, but also provides qualitative explanation of how generalization is affected by the structure of the renormalization matrix in different tasks.

**Effect of regularization strengths on generalization performance:** Similar to the dependence on $N$, generalization also exhibits different behavior as a function of the regularization parameter $\sigma$ in different parameter regimes, with contributions from both the bias and the variance. The dependence of error bias on $\sigma$ also arises due to the matrix nature of the renormalization. In Fig.3 , we show parameter regimes where the bias can increase (Fig.3(a-c)) or decrease (Fig.3(d-f)) with $\sigma$ on MNIST dataset [19] (Appendix C.3 ). Although the dependence on $\sigma$ is complicated and diverse, and there lacks a general rule for when the qualitative behavior changes, we found that our theory accurately captures the qualitative behavior of results obtained from GD (Appendix B.3 Fig.3). In both regimes the variance increases with $\sigma$ as the solution space expands for a weaker regularization. Specifically in Fig.3 (d-f), due to the increasing variance (e) and decreasing bias (d), there is a minimum error rate ((f), Appendix A.3 Eq.50 for how error rate is calculated from the mean and variance of the predictor) at intermediate $\sigma$, indicating an optimal level of regularization strength as opposed to linear networks [4], where strong regularization ($\sigma = 0$) always results in optimal generalization

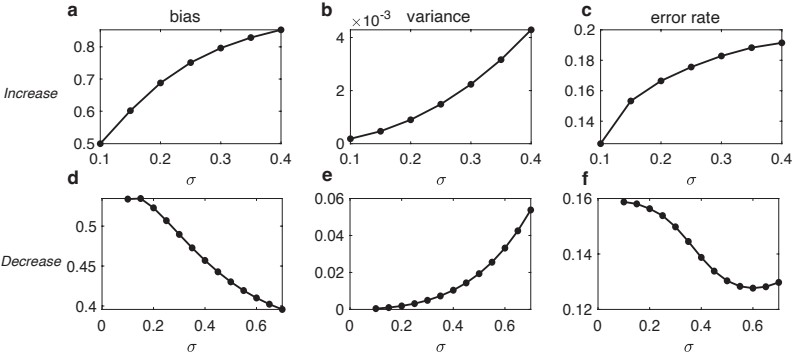

Figure 3: Generalization as a function of $\sigma$ for GGDLNs trained on MNIST dataset predicted by our theory. (**a-c**) Bias (a), variance (b) and error rate (c) increase as a function of $\sigma$ . (**b-f**) Bias decreases as a function of $\sigma$ while variance increases, leading to an optimal $\sigma$ with minimum error rate.

**GGDLNs with different choices of gatings achieve comparable performance to ReLU networks:** The nonlinear operation of the gatings enables the network to learn nonlinear tasks. In Fig.4, we show that although the gatings are fixed during training, the network achieves comparable performance as a fully trained nonlinear (ReLU) network with the same hidden layer width for classifying even and odd digits in MNIST data when $M$ is sufficiently large (over-parameterization does not lead to over-fitting

here, as shown also in other nonlinear networks [27, 28], possibly due to the explicit $L_2$ prior). Furthermore, although the gatings are fixed during the supervised training of the GGDLN, they can be cleverly chosen to improve generalization performance. To demonstrate this strategy, we compared two different choices of gatings. *Random gatings* take the form $g_m(\mathbf{x}) = \Theta(\frac{1}{\sqrt{N_0}}\mathbf{V}_m^\top\mathbf{x} - b)$, where $\mathbf{V}_m$ is a $N_0$-dimensional random vector with standard Gaussian i.i.d. elements, $b$ is a scalar threshold, and $\Theta(x)$ is the heaviside step function. The *pretrained gatings* are trained on the *unlabelled* training dataset with unsupervised soft k-means clustering, such that the $m$-th gating $g_m(\mathbf{x})$ outputs the probability of assigning data $\mathbf{x}$ to the $m$-th cluster (Appendix C.3). As shown in Fig. 4, for pretrained gatings, generalization performance improves with $M$ much faster compared to random gatings, and approaches the performance of ReLU network at a smaller $M$. Our theory (Fig. 4) and numerical results of GD dynamics (Appendix B.3 Fig. 4) agree qualitatively well. The result shows that GGDLNs can still achieve competitive performance on nonlinear tasks while remaining theoretically amenable.

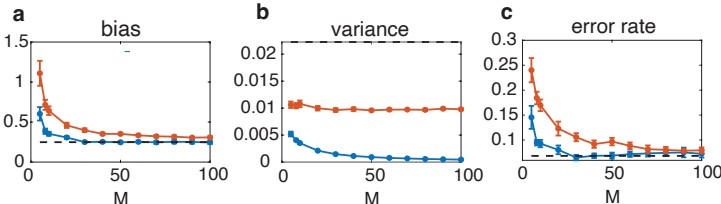

Figure 4: Dependence of generalization on M for GGDLNs trained on MNIST dataset predicted by our theory. Bias (a), variance (b) and error rate (c) as a function of M for random (red lines) and pretrained gatings (blue lines), and ReLU network with the same width (black dashed lines).

## 4.2  Kernel shape renormalization in deeper networks

We now consider the effect of the matrix renormalization on GGDLNs with more layers. We begin by analyzing the renormalization effect on the *shape* of the kernel in deep architectures. It is well known that the GP kernel of many nonlinear networks flattens (the kernel function goes to a constant) as network depth increases [2], ultimately losing information about the input and degrading generalization performance. Here we show that kernel shape renormalization slows down flattening of kernels by incorporating data relevant information into the learned weights.

To study the *shape* of the kernel independent of kernel magnitude, we define the normalized kernel $\mathcal{K}_L(\mathbf{x}, \mathbf{y}) = \frac{\tilde{K}_L(\mathbf{x},\mathbf{y})}{\tilde{K}_L(\mathbf{x},\mathbf{x})^{1/2}\tilde{K}_L(\mathbf{y},\mathbf{y})^{1/2}}$, where $\tilde{K}_L(\mathbf{x}, \mathbf{y})$ denotes the renormalized kernel for GGDLN with $L$ hidden layers. This normalized kernel measures the cosine of the vectors $\mathbf{x}$ and $\mathbf{y}$ with generalized inner product defined by the kernel $\tilde{K}_L(\mathbf{x}, \mathbf{y})$, and therefore $\mathcal{K}_L(\mathbf{x}, \mathbf{y}) \in [-1, 1]$. For the GP kernel of GGDLNs, we have $\mathcal{K}_L(\mathbf{x}, \mathbf{y}) = \cos(\mathbf{g}(\mathbf{x}), \mathbf{g}(\mathbf{y}))^L \cos(\mathbf{x}, \mathbf{y})$. While $\mathcal{K}_L$ depends on the specific choice of gatings in general, in the special case of *random gatings* with zero threshold $g_m(\mathbf{x}) = \Theta(\frac{1}{\sqrt{N_0}}\mathbf{V}_m^\top\mathbf{x})$ and the number of gatings $M \to \infty$, we can write $\mathcal{K}_L$ analytically as a function of the angle $\theta$ between input vectors $x$ and $y$, given by $\mathcal{K}_L(\theta) = (\frac{\pi-\theta}{\pi})^L \cos(\theta), \theta \in [-\pi, \pi]$. Thus, as $L \to \infty$, $\mathcal{K}_L(\theta)$ shrinks to zero except for $\theta = 0$. This 'flattening' effect reflects the loss of information in deep networks, as pairs of inputs with different similarities now all have hidden representations that are orthogonal. The effect also empirically holds true for networks with finite $M$ (see Appendix B.4).

In Fig. 5 ,we study the effect of kernel renormalization on the 'flattening' effect of deep GGDLNs. As shown in Fig. 5 (a)-(c), the elements of the renormalized kernel shrink to zero at a much slower rate compared to the GP kernel. (Note that unlike the variance, the bias is affected only by shape changes, but not by changes in the amplitude of the kernel, in Fig. 5(d) we plot only the bias contribution to the generalization.) While mitigating the flattening of the GP kernel is a general feature of our renormalized kernel for different parameters, its effect on the generalization performance (especially the bias) may be different for different network parameters. In the specific example in Fig. 5, finite width networks with a less 'flattened' renormalized kernel achieve better performance than the GP limit. Both the GP limit and the finite width networks have optimal performance at $L = 2$ in this example.

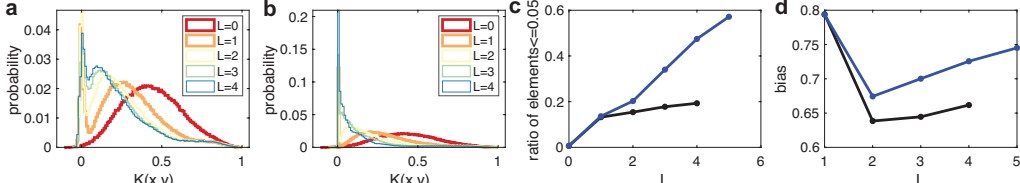

Figure 5: Shape renormalization slows down flattening of kernels in deep networks. (**a-b**) Distribution of kernel elements $\mathcal{K}_L(\mathbf{x}, \mathbf{y})$ for the renormalized kernel (a) and GP kernel (b) for different network depth $L$. (**c**) Ratio of kernel elements smaller or equal to 0.05 increases faster for GP kernel (blue line) compared to the renormalized kernel (black line), the renormalization slows down the rate at which elements in the GP kernel shrink to zero as a function of $L$. (**d**) The bias contribution to the generalization first decreases then increases as a function of $L$ due to the flattening of the kernel (blue line). Finite width network with renormalized kernel performs better for $L > 1$ in this parameter regime (black line). See Appendix C.3 for detailed parameters.

## 5 GGDLNs for multiple tasks

In this section, we apply our theory to investigate the ability of GGDLNs to perform multiple tasks. We consider two different scenarios below. First, different tasks require the network to learn input-output mappings on input data with different statistics. This scenario corresponds to real life situations where the training data distribution is non-stationary. The tasks can be separated without any additional top-down information. In this case, the gatings are bottom-up, and are functions of the input data only. In the second case, different tasks give conflicting labels for the same inputs, corresponding to the situation where performing the two tasks require additional top-down contextual information, and the information can be incorporated into the gating units in GGDLNs. In both scenarios, when the gatings are fixed and we modulate the de-correlation by changing network width and thus the strength of the kernel renormalization, we find that de-correlation between tasks leads to better generalization performance.

### 5.1 Bottom-up gating units

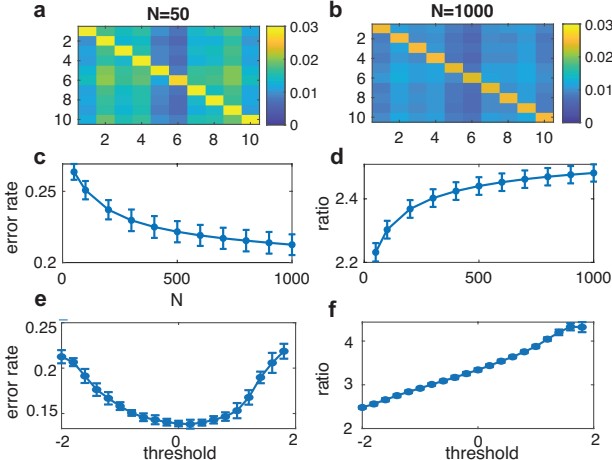

Figure 6: GGDLNs with bottom-up gating units learning multiple tasks trained on permuted MNIST. (**a-b**) Task-task correlation matrix C for $N = 50$ and $N = 1000$, different permutations are more decorrelated for larger N. (**c**) Error rate decreases as a function of N due to the decorrelation. (**d**) Ratio of the average amplitude of diagonal elements versus off-diagonal elements in C increases as a function of N. (**e**) Error rate first decreases then increases as a function of gating threshold. (**f**) Decorrelation increases as a function of gating threshold.

First we consider learning different tasks defined by vastly different input statistics with bottom-up gatings, using permuted MNIST as an example. Previous works have shown that GLNs mitigate catastrophic forgetting when sequentially trained on permuted MNIST [8]. While our theory does not address directly the dynamics of sequential learning, we aim to shed light on this question by asking how the two tasks interfere with each other when they are learned simultaneously.

We introduce a measure of *inter-task interference* by noting that after learning the mean predictor on a new data $\mathbf{x}$ , Eq.8, is a linear combination of the output labels $Y^\mu$ of all the training data,

and the coefficient of this linear combination, is given by the $\mu$-th coefficient of $\tilde{\mathbf{k}}(\mathbf{x})^\top \tilde{\mathbf{K}}^{-1}$. Thus, we define a task-task correlation matrix, via $C_{pq} = \sum_{\mu=1}^{P} \sum_{\gamma=1}^{P_t} |\tilde{\mathbf{k}}^T \tilde{\mathbf{K}}^{-1}|_{p\gamma,q\mu} (p, q = 1, \cdots, n)$, where we assume there are $P$ training examples and $P_t$ test data for each task, with a total of $n$ tasks. The amplitude of each element $C_{pq}$ measures how much *training data of task $q$* contribute to the prediction on the *test data of task $p$*. Stronger diagonal elements indicates that the network separates the processing of data of different tasks (Fig. 6(a)-(b)). As we show in Fig. 6, we can tune the relative strength of the diagonal elements of $\mathbf{C}$ smoothly by changing the network width (Fig. 6(a)-(d)) or by changing the threshold of the gating (Fig. 6 (e)-(f)). In the case where the gatings are fixed and the network width is changed, an increase in the strength of the diagonal elements (Fig. 6(d)) results in better generalization (Fig. 6(c)), indicating that the network generalizes better by processing data of different tasks separately through the gating units. However, in the case where we change the activation of the gatings by adjusting the threshold, although different tasks are more de-correlated when the threshold is large due to a set of less overlapping gatings activated for each task, generalization error first decreases and then increases again. This is because for large threshold the sparsity of the gatings activated for each task limits the nonlinearity of the network, and therefore the generalization performance on this nonlinear task.

## 5.2   Combined top-down and bottom-up gating units

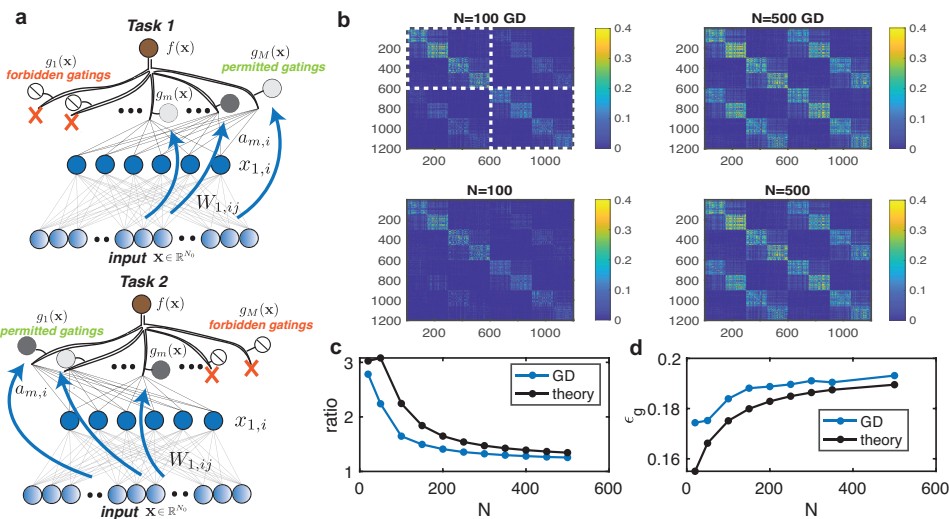

Figure 7: Kernel renormalization de-correlates different tasks defined by different labels on the same inputs. (**a**) GGDLNs performing two tasks using combined top-down and bottom-up task signal. (**b**) Top: Renormalized kernel calculated with Eq. 7 from GD dynamics. Bottom: Renormalized kernel theory. (**c**) Ratio of the magnitude of diagonal (blocks with white dashed lines in (b)) versus off diagonal blocks decreases as a function of $N$. (**d**) Generalization error increases with N.

We now consider learning two tasks that provide conflicting labels on the same input data. The gating units combine both top-down task signal which informs the system of which task to perform for a given input, and bottom-up signals which, as before, depend on the input. In different tasks, different sets of gatings are *permitted or forbidden* depending on the top-down signal, then the states of the permitted gatings are further determined as a function of the input $\mathbf{x}$, while the forbidden gatings are set to $0$, and the corresponding dendritic branches do not connect to the previous layer neurons (Fig. 7(a)) in this task. For a single hidden layer network, with a similar argument as in Section 2, it is straightforward to show that the number of different tasks that can be memorized is given by $n \leq M$ and the number of training examples for each task needs to satisfy $P \leq N_0 M_p$, where $M_p$ is the number of permitted gating units in each task. In the limiting case where a set of non-overlapping gating units are permitted in each of the $n$ tasks, the network is equivalent to $n$ sub-networks, each independently performing one task. In this case $M_p$ is limited by $M/n$, which in turn limits the capacity and the effective input-output nonlinearity for each independent task. We consider the case where the permitted gatings are chosen randomly for each task and are therefore in

general overlapping across tasks. We then investigate how learning modifies the correlation induced by the overlapping gatings through the renormalization matrix. As an example we consider training on permuted and un-permuted MNIST digits of 0 and 1's. One task is to classify the two digits in both permuted and un-permuted data, and the second task is to separate the permuted digits (both 0 and 1) from the un-permuted digits. The labels of the two tasks are uncorrelated, while the permitted gatings of the two tasks are partially overlapping. In this case the renormalized kernel $\tilde{\mathbf{K}}_1$ can be written as $\tilde{K}_{p\mu,q\nu} = (\frac{1}{M}\mathbf{g}^p(\mathbf{x}^\mu)^\top \mathbf{U}_1 \mathbf{g}^q(\mathbf{x}^\nu))\frac{\sigma^2}{N_0}\mathbf{x}^{\mu\top}\mathbf{x}^\nu$. Here $p,q \in \{1,2\}$ are the task indices, and $\mu,\nu = 1,\cdots,P$ are the input indices. The kernel is therefore $2P \times 2P$ as shown in Fig. 7(b) ($P = 600$); the diagonal blocks (white dashed lines) correspond to kernels of task 1 and task 2, while the off diagonal blocks correspond to the cross kernels. In Fig. 7(b) bottom, we show the renormalized kernel with the renormalization matrix $\mathbf{U}_1$ calculated by solving Eq. 6. Similar results are achieved by by numerically estimating Eq. 7 with readout weights obtained from GD dynamics (Fig. 7(b) top).

The results demonstrate that stronger kernel renormalization achieved in narrower networks suppresses more strongly the correlation between tasks, reflected by the weaker off-diagonal blocks in Fig. 7(b). A decreasing ratio between the average amplitudes of the diagonal and off-diagonal blocks shows that the de-correlation effect diminishes for large $N$, leading to increasing generalization error with $N$(Fig. 7(c & d)).

## 6 Discussion

In this work, we proposed a novel gating network architecture, the GGDLN, amenable to theoretical analysis of the network expressivity and generalization performance. The predictor statistics of GGDLNs can be expressed in terms of kernels that undergo shape renormalization, resulting diverse behavior of the bias as a function of various network parameters. This renormalization slows down the flattening of the GP kernel in deep networks, suggesting that the loss of input information as $L$ increases may be prevented in finite-width nonlinear networks. We also investigate the capability of GGDLNs to perform multiple tasks. While our theory is an exact description of the posterior of weight distribution induced by Langevin dynamics in Bayesian learning, it provides surprisingly well qualitative agreement with results obtained with GD dynamics for not only the generalization but also the kernel representation with matrix renormalization, largely extending its applicability. There are several limitations of our work. Our mean-field analysis is accurate in the 'finite-width' thermodynamic limit where both $P$ and $N$ go to infinity, but $M$ and $L$ remain finite. In practice, the size of the renormalization matrix increases as $M^L$, hence for some moderate $M$, as $L$ increases, any large but finite $N$ might eventually get the network outside the above thermodynamic regime. The theory also focuses on the equilibrium distribution induced by learning and does not address important questions related to the learning dynamics. Finally, although we have shown qualitative correspondence of the GGDLN properties and standard DNNs with local nonlinearity, as ReLU, a full theory of the thermodynamic limit of DNNs with local nonlinearity is still an open challenge.

While our theory currently addresses learning in GGDLNs using a *global* cost function, exploring the possibility of extending the formalization of the equilibrium distribution to characterize local learning dynamics is an ongoing work. Recent works have shown that multilayer perceptrons (MLPs) with learned gatings that implements spatial attention have surprisingly good performance on Natural Language Processing (NLP) and computer vision [29]. Extension of our theory to learnable gatings that implements attention mechanisms remains to be explored. Furthermore, incorporating convolutional architecture [30, 31, 32, 33, 34] into our GGDLNs and using the gating units to encode context-dependent modification of different feature maps is an interesting direction related to the fast-developing research topic of visual question-answering (VQA) [35, 36, 37] , where answering different questions about the same image is similar to performing multiple tasks in different contexts with different labels on the same dataset, as we discussed in Section 5.2. We leave these exciting research directions for future work.

## Acknowledgement

We thank the anonymous reviewers for their helpful comments. This research is supported by the Swartz Foundation, the NIH grant from the NINDS (No. 1U19NS104653), and the Gatsby Charitable Foundation. We acknowledge the support of a generous gift from Amazon. This paper is dedicated to the memory of Mrs. Lily Safra, a great supporter of brain research.

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
