# OpenReview forum: "Globally Gated Deep Linear Networks"
_NeurIPS.cc/2022/Conference — NeurIPS 2022 Accept_

### Official Review · Reviewer_PKE5 · 2022-07-08

**Rating:** 7
**Confidence:** 3
**Soundness:** 3 good
**Presentation:** 3 good
**Contribution:** 3 good

**Summary:**

The authors propose a new architecture based on Gated Linear Networks (GLN) called Globally Gated Deep Linear Network (GGDLN). They derive equation for the generalization properties and describe the architecture theoretically. They present also several experiments comparing GGDLN and the same architecture learned by Gradient Descent.

**Questions:**

The formalism in Fig1 is different from the one used in the equations.
For example $x_i^L$ in Figure and $x_{L, i}$ in the equation

The Fig1(a) should be improved because it is not so clear. Some suggestions:
- the number of neurons at layer L and L-1 are not explicit (it should be N for all layers right?)
- it is not clear that $W_{i0}^m$ ...  $W_{iN}^m$ are: the connections between the m-th dendrite of i-th neuron and the neurons 0...N in the previous layer. Maybe the other connections could be depicted with dashed line and focus only on the part important for the computation in Eq.1.
- it is necessary to define also the layer. In the equation (1), in fact, the authors use $W_{L,ij}^m$.
- $x$ is a vector I suggest to use the proper style (e.g. bold, $\mathbf{x}$) in the figure and throughout the paper.

Since, based on my understanding, $a_{i,m}$ is the i-th element of $\mathbf{a_m}$, maybe is more coherent with other approaches in eq.1 to use $a_{m,i}$. However the authors should use a formalism and be coherent throughout the paper

Equation (1) what are the limits of j in the summation?

row 79 --> I didn't understand the formalism. What are $m_1$, $m_2$, etc. What is the meaning of the summation? Maybe I understood the sense but the formalism is not clear neither introduced before.

If I understood well, the parameters are learned maximizing the posterior probability in (3) [as described rows 143-145] but in rows 87-101 is not clearly stated this.

row 118 and Appendix C4 --> Is the network defined in Figure 1 trained using GD as comparison?

What do you mean as ReLU Teacher task and what do the student network is supposed to do?

The caption of Figure 2 is not clear. I suggest to use bold style for (a), (b), (c), and (d) when the successive text is related to them because, as now, the caption is confusing. I suggest to review all the captions because this problem is common.

row 163-174 : What do you mean with localized receptive fields for the gatings? I didn't understand this part. Could you clarify it?

I didn't understand the rationale behind the comparison of a learning using the proposed approach and the GD learning approach.
Why does these modalities should behave the same? Also the authors said that "While we do not expect our theory to agree with GD dynamics quantitatively, as we will show, our theory makes accurate qualitative predictions for GD dynamics in all examples throughout this paper.".
What do author mean with qualitative? The shapes are different and maybe it is not a big deal because the modalities are different. Instead it is important to demonstrate that the proposed approach has a good performance on several tasks (not necessary outperform GD but that accomplish a given task as described in rows 185-201) but the rationale behind the experimental setting is not clear to me.

Section 5 looks like a separated part that could deserve a paper itself. Maybe I remove it and devote more space to clarifications, descriptions, assumption, discussion, etc.

The Discussion section looks like a Conclusion paragraph.

**Limitations:**

The authors describe more the next perspectives that limitations.

**Strengths And Weaknesses:**

The main strength is the attempt to analyze the Gated Deep Linear Networks on the theoretical basis. Moreover the authors have made a considerable effort also to analyse the architecture by different perspectives.

My main concern is that the work is too dense. Nine pages looks too few to contain all the material authors want to present making the reading process difficult (also because the introductory figures are not so good as necessary). For this reason some details are missing, discussion is very short and some clarifications are missing.

---

> ### Author Response · Authors · 2022-07-31
> **Response to reviewer PKE5**
>
> Response to questions:
> 1. We apologize for the inconsistency in the notations and the formalism. We have gone through this paper and fixed them in the revised manuscript. We have also modified the figure captions (bold letters) as the reviewer suggested.
> 2. Row 79: This is simply another way to write Eq.1. We have added more detailed explanation of what the new indices are and what $\bf{x}^{\rm{eff}}$ and $\bf{W}^{\rm{eff}}$ are. We are expressing the input-output relation in this way to explicitly show that the output of the network is essentially linear in the effective input $\bf{x}^{\rm{eff}}$, which gives us a simple argument for the expression for the memory capacity of the network.
> 3. Here we are considering the capacity, which refers to the maximum number of random (or generic) input-output examples, for which there exists a set of network parameters which yields zero training error (MSE). It is irrespective of the learning algorithm, which is why the Bayesian formulation comes after the capacity section. To further clarify, we are not ‘maximizing the posterior’. Instead, we consider the full distribution and derive generalization properties given by the statistics of the parameters (Section 2). For instance, the mean predictor is computed by averaging with respect to the posterior distribution (not by computing at the maximal posterior).
> 4. Row 118: We do show comparison with GD for various example tasks on the network structure in Fig.1(a) throughout the paper. In Fig.1(b-d) specifically, we are making a statement about the capacity of the network and the drastic change of generalization behavior around the interpolation threshold. As this phenomenon is not unique to our ‘finite width kernel renormalization’ theory, so we show it for the infinite width limit, before introducing our treatment of the finite width networks. However, we do show the same double descent phenomenon in both the finite width theory and the GD dynamics of this architecture in SI Fig.1 top.
> 5. ReLU teacher task: The teacher network is a neural network with one hidden layer and ReLU nonlinearity. The student GGDLN is required to learn the input-output association generated by the teacher. The teacher-student setup is commonly used for evaluating and understanding neural network performance [1,2,3].
> 6. Localized receptive field: As illustrated and explained in Figure 2(a) and its corresponding caption, it means that each gating unit is only connected to a subset of the input dimensions.
> 7. Rationale behind comparison with GD: Our aim is not to propose a new architecture with superior performance, but to generate theoretical insights into the generalization performance of GGDLN trained by supervised learning. While our posterior does not correspond exactly to a simple GD dynamic, it is interesting to ask to what extent the predicted behaviors are also exhibited by GD dynamics of the same network architectures. Our simulations with GD provide convincing evidence that indeed the Bayesian theory’s predictions are qualitatively similar to those observed in the traditional GD learning, where the strength of the L2 regularizer in the Bayesian theory is mapped into the inverse variance of the Gaussian used to generate the initial weights in the GD dynamics. By ‘qualitative’ we mean that with the same network setup, the dependence on various network parameters are the same for the theory and GD simulations (Several examples: In Fig.2&3, they either both increase or both decrease with N or $\sigma$. In Fig.4, they both decrease with M and approach the performance of ReLU networks. In Fig.7, for both theory and GD simulation, the off-diagonal (red) part of the renormalized kernel becomes less pronounced for small N compared to GP).
> 8. Discussion: For the camera-ready version we would like to add the following paragraph to the Discussion section to address our limitations more thoroughly
> ‘There are several limitations of our work. Our mean-field analysis is accurate in the ‘finite-width’ thermodynamic limit where both P and N go to infinity, but M and L remain finite. In practice, the size of the renormalization matrix increases exponentially with L, as $M^L$, hence as L increases, any large but finite N might eventually get the network outside the above thermodynamic regime. The theory also focuses on the equilibrium distribution induced by learning and does not address important questions related to the learning dynamics. Finally, although we have shown qualitative correspondence of the GGDLN properties and standard DNNs with local nonlinearity, as ReLU, a full theory of the thermodynamic limit of DNNs with local nonlinearity is still an open challenge’.
> [1] Wang et al, IEEE Transactions on Pattern Analysis and Machine Intelligence (2021).
> [2] Watanabe et al, IEEE International Conference on Acoustics, Speech and Signal Processing (ICASSP), 2017.
> [3] Advani et al, Neural Networks 132 (2020): 428-446.

---

> > ### Comment · Reviewer_PKE5 · 2022-08-07
> > **Response**
> >
> > I thank the authors for their answers and clarifications.
> >
> > I advise to authors to carefully double check the equations and formalism because it looks like there are still some little things to fix.
> >
> > Moreover, I think that some clarifications authors gave me could be useful for a generic reader and I suggest them to include in the paper or in appendix (e.g., the rationale behind the comparison with GD, point 3, etc)
> > Based on the answers and the revised paper, I decided to increase my initial rating.

---

### Official Review · Reviewer_Psan · 2022-07-13

**Rating:** 7
**Confidence:** 2
**Soundness:** 4 excellent
**Presentation:** 2 fair
**Contribution:** 3 good

**Summary:**

This paper presents theoretical insights that build upon the recently proposed Gated Linear Networks (GLNs). First, a simplified version of (GLNs) called Globally Gated Deep Linear Networks (GGDLNs) are proposed. The motivation for the proposal is to construct a model that can serve as a useful model of learning in general neural networks by being practical enough, yet amenable to theoretical analysis. The proposed model enables the authors to obtain useful theoretical characterizations of the memory capacity and generalization behavior for GGDLNs, which are confirmed via simulations.

**Questions:**

In Eq. 1, shouldn't there be a summation over $M$ dendrites when computing neuron outputs for the intermediate layers? How are the $M$ dendritic activations used to compute the $N$ neuron outputs otherwise?

**Limitations:**

Yes, the limitations are discussed in Sec. 6.

**Strengths And Weaknesses:**

I should note that I am not familiar with the prior work in this area, and thus my opinions are only based on a few readings of this paper as a newcomer. I can not ascertain if all relevant prior work has been properly credited, and can not verify that the method of Backpropagating Kernel Renormalization (Li & Sompolinsky, 2021) has been applied correctly to the proposed model.
Nevertheless, the authors make a good case that GGDLNs are a more interesting model to study compared to deep linear networks studied in recent work, exhibiting generalization behavior that is closer to neural networks used in practice (Sec. 4). It is also shown that there is remarkable qualitative agreement between the paper's theoretical predictions regarding generalization and results obtained using gradient descent, even though technically the theory applies to posterior predictions obtained using Langevin Dynamics. This again indicates that GGDLNs are useful objects to study for predicting the behavior of neural networks in practice.

---

> ### Author Response · Authors · 2022-07-31
> **Response to reviewer Psan**
>
> Thank you for your careful review and support of our work. There should indeed be a summation over the dendritic index $m$ from $1$ to $M$, we have fixed the typo. Thank you for pointing this out.

---

### Official Review · Reviewer_k3LQ · 2022-07-14

**Rating:** 7
**Confidence:** 2
**Soundness:** 4 excellent
**Presentation:** 3 good
**Contribution:** 3 good

**Summary:**

The authors propose Globally Gated Deep Linear Networks (GGDLNs), which is a variant of Gated Linear Networks (GLNs) that is more amenable to theoretical analysis. The paper presents a wide breadth of theoretical results, which are additionally supported by simulations.  These include bias & variance analysis for single-layer networks, kernel normalization analysis for multi-layer networks, and multi-task learning.

**Questions:**

1- Is it possible to apply the kernel shape renormalisation theory to GGDLNs with local learning (like in GLNs) instead of backpropagation?

2- Are GGDLNs universal function approximators?

3- Have the authors observed the "double descent" with GGDLNs in the simulations?

4- Figure 4 shows that decreasing generalisation error with M. However, as the capacity grows quickly with M, we would expect to see overfitting. Why is this not the case?

5- Can the authors explain what the flattening effect mean in practice? And is it observed in, eg, ReLu networks?

6- Error rate decreases as a function of the threshold in Figure 6.b. For even larger thresholds, eg (5, 10) we would expect to see the error rate to increase again, because the hyperplanes will not be able to separate the data. Is this the case in your simulations? If so, it would be insightful to widen the range of the x-axis.

**Limitations:**

Limitations are adequately addressed.

**Strengths And Weaknesses:**

Backpropagating Kernel Renormalization (BKR) has been applied for analysing multi-layer linear networks. One of the contributions of this work is to extend the BKR analysis to multi-layer non-linear networks.  This non-linearity is achieved via gating as opposed to static non-linearities such as ReLu functions, which makes the analysis tractable. This transition from linear functions to non-linear functions is a significant theoretical progress.

The theoretical analysis is sound and insightful. Most results are intuitive/expected; however, I am confused by some of them, as I explain in Questions 3-6. I will update my evaluation after understanding some of the results better.

The pretrained gating idea is novel and shows promise for GLN-like architectures.

Perhaps the biggest weakness is that the analysis is limited to the GGDLNs, and do not directly apply to popular architectures such as ReLu MLPs. However, analysis of non-linear networks is extreme difficult. So any progress is valuable. As the authors note, the analysis techniques developed in the paper might be applicable to more popular architectures in the future.

The paper is written and presented clearly. However, it can be quite dense in some places, which is expected given the quantity of results.

There are a few minor writing issues:
- The text could benefit from a spell-check ("processisng",  "analayzing" etc).
- The x-axis label in Figure 1 must be P instead of M I believe.
- Panel label d is missing in Figure 7. See caption "(d) Generalization error increases with..."

---

> ### Author Response · Authors · 2022-07-31
> **Response to reviewer k3LQ**
>
> Weaknesses:
> We have gone over the paper carefully and fixed the typos and writing issues. However, the x-axis label in Figure 1 is indeed $M$, here we fixed $P$ and vary $M$, which changes the network capacity. For large $M$ (above the dashed line) the network is below capacity and reaches zero training error, for small M the network is above capacity, and the training error is nonzero. Here we demonstrate a double descent behavior with respect to increasing the network complexity, in our case M, reminiscent of the double descent phenomenon in other nonlinear networks.
>
> Response to Questions:
> 1. Our present theory is not directly applicable to the local learning rule for the following reason. In the local learning dynamics, the update of the $L$-th layer weights $W_L$ depends only on the activation of the current and the previous (i.e., $L-1$-th layer neurons). This type of update rule dynamics does not directly correspond to minimizing a simple energy function, which is essential to our formulation of Gibbs distribution of the weights. However, in on-going work we explore possibilities to formalize an appropriate equilibrium distribution for local learning dynamics. We comment on it in the revised manuscript.
> 2. Intuitively, it is to be expected on general grounds, that if $M$ is sufficiently large and the gating are chosen at random, they offer sufficiently rich infinitely large nonlinear basis functions to allow for universal approximations. Of course, a detailed rigorous proof needs to specify the space of functions as well as the structure of the gating (along similar lines to Ref [1]). The real interesting and difficult problem is to provide a bound of the rate of convergence of such approximation with $M$. This interesting topic remains to be explored in future work.
> 3. Yes, we have indeed observed a double descent, but as a function of $M$ instead of network width $N$. This is expected because for double descent, the generalization error typically peaks at the interpolation threshold, which in standard nonlinear networks (e.g. ReLU nonlinearity) is determined by the network width $N$, while in GGDLNs capacity is controlled (in large $N$) by $M$. Figure 1(d) shows the GP theory, where generalization error peaks at the interpolation threshold and then decreases again, corresponding to the double descent phenomenon. In SI Figure 1 top, we show our simulation results for finite width GGDLNs with GD, which exhibits similar double descent phenomenon.
> 4. This is related to the previous question about the double descent phenomenon. It is well known that many networks in the over-parametrization regime do not overfit, and their performance keeps improving as the number of parameters increases. This improvement has been explained by the ‘small norm’ inductive bias of the common training algorithms for neural networks, either implicitly in SGD and appropriate initial weights or explicitly by $L_2$ regularizers or early stopping, ([2]). In our case we have an explicit $L_2$ regularizer which biases the weight space towards the solutions with smaller $L_2$ norms.
> 5. The flattening effect happens in the GP limit for several common types of nonlinear networks (including ReLU and tanh nonlinearity) ([5]). For example in ReLU networks as studied in [3], the kernel function $K_{GP}(x,y)$ (normalized by its diagonal values) is actually a function of the angle between the two input vectors x and y. As shown in Figure 6 in [3], as depth increases the kernel function becomes a constant, implying the gradual loss of information about the input structure in deep networks, as the (normalized) similarity between the hidden layer representations of any pair of inputs becomes constant, irrespective of their input similarity. Here we show a similar but not exactly identical ‘flattening’ effect in GGDLNs. Here the (normalized) GP kernel converges to zero for any two unparallel input vectors (i.e., the corresponding representation pairs are orthogonal). This effect still reflects the loss of information, as pairs of inputs with different similarities now all have hidden representations that are orthogonal. The practical implication of this flattening in both ReLU networks and GGDLNs is that the generalization performance is optimized at some finite depth.
> 6. Yes, it is indeed the case. We have widened the range of the x-axis in Figure 6 (e)&(f).
> [1] Veness, Joel, et al. "Online learning with gated linear networks." arXiv preprint arXiv:1712.01897 (2017).
> [2] Belkin, Mikhail, et al. "Reconciling modern machine-learning practice and the classical bias–variance trade-off." Proceedings of the National Academy of Sciences 116.32 (2019): 15849-15854.
> [3] Lee, Jaehoon, et al. "Deep neural networks as gaussian processes." arXiv preprint arXiv:1711.00165 (2017).

---

> > ### Comment · Reviewer_k3LQ · 2022-08-07
> > **Thank you**
> >
> > I thank the authors for the insightful comments. I am further reassured by the soundness of the results. My recommendation remains unchanged.

---

### Meta-Review · Area_Chair_PH2c · 2022-08-27

**Recommendation:** Accept
**Confidence:** Less certain

**Metareview:**

Three reviewers recommend accept. The reviewers praised the significant extensions of previous works, the novelty of the ideas, and found the theoretical analysis sound and insightful. The author responses to initial feedback were found to be insightful and reassured the reviewers about their recommendations. Hence I am recommending accept. I encourage the authors to take the reviewers feedback carefully into consideration when preparing the final manuscript and to work on the items promised in the discussion period.

**Award:**

No

---

### Decision · Program_Chairs · 2022-09-14

Accept